Comprehensive classification of the plant non-specific lipid transfer protein superfamily towards its sequence–structure–function analysis

Fleury Cécile 1
Gracy Jérôme 2
Gautier Marie-Françoise 1
Pons Jean-Luc 2
Dufayard Jean-François 3
Labesse Gilles 2
http://orcid.org/0000-0001-8153-276X Ruiz Manuel 3
http://orcid.org/0000-0003-4234-1172 de Lamotte Frédéric 1 frederic.de-lamotte@inra.fr
1 UMR AGAP, INRA , Montpellier , France
2 CBS, CNRS Univ Montpellier INSERM , Montpellier , France
3 UMR AGAP, CIRAD , Montpellier , France
Sotelo-Mundo Rogerio
Electronic publication date: 2019 Aug 14
Publication date: 2019
Volume: 7
Electronic Location ID: e7504
Received 2019 Apr 25; Accepted 2019 Jul 17
Copyright: © 2019 Fleury et al.
Copyright year: 2019
Copyright holder: Fleury et al.
License: This is an open access article distributed under the terms of the Creative Commons Attribution License, which permits unrestricted use, distribution, reproduction and adaptation in any medium and for any purpose provided that it is properly attributed. For attribution, the original author(s), title, publication source (PeerJ) and either DOI or URL of the article must be cited.
License URL: https://creativecommons.org/licenses/by/4.0/

Keywords: nsLTP, Plant, Phylogeny, Molecular modeling, Structure–function relationships, Multigenic family, Functional annotation, Homology modeling

Funding: French National Research Agency (ANR Genoplante) ANR- 08-GENO118 and ANR-10-BINF-03-04 This work was supported by the French National Research Agency (ANR Genoplante) (grant ANR- 08-GENO118) and (ANR-10-BINF-03-04). The funders had no role in study design, data collection and analysis, decision to publish, or preparation of the manuscript.

==============================
Background

Non-specific Lipid Transfer Proteins (nsLTPs) are widely distributed in the plant kingdom and constitute a superfamily of related proteins. Several hundreds of different nsLTP sequences—and counting—have been characterized so far, but their biological functions remain unclear. It has been clear for years that they present a certain interest for agronomic and nutritional issues. Deciphering their functions means collecting and analyzing a variety of data from gene sequence to protein structure, from cellular localization to the physiological role. As a huge and growing number of new protein sequences are available nowadays, extracting meaningful knowledge from sequence–structure–function relationships calls for the development of new tools and approaches. As nsLTPs show high evolutionary divergence, but a conserved common right handed superhelix structural fold, and as they are involved in a large number of key roles in plant development and defense, they are a stimulating case study for validating such an approach.

Methods

In this study, we comprehensively investigated 797 nsLTP protein sequences, including a phylogenetic analysis on canonical protein sequences, three-dimensional structure modeling and functional annotation using several well-established bioinformatics programs. Additionally, two integrative methodologies using original tools were developed. The first was a new method for the detection of (i) conserved amino acid residues involved in structure stabilization and (ii) residues potentially involved in ligand interaction. The second was a structure–function classification based on the evolutionary trace display method using a new tree visualization interface. We also present a new tool for visualizing phylogenetic trees.

Results

Following this new protocol, an updated classification of the nsLTP superfamily was established and a new functional hypothesis for key residues is suggested. Lastly, this work allows a better representation of the diversity of plant nsLTPs in terms of sequence, structure and function.

Introduction

Since the work of Kader, Julienne & Vergnolle (1984) and Kader (1996), numerous proteins capable of transferring lipids have been annotated as non-specific lipid transfer proteins (nsLTPs). Their primary sequences are characterized by a conserved 8-Cysteine Motif (8CM) (C-Xn-C-Xn-CC-Xn-CXC-Xn-C-Xn-C), which plays an important role in their structural scaffold (José-Estanyol, Gomis-Rüth & Puigdomènech, 2004). Based on their molecular masses, plant nsLTPs were first separated into two types: type I (nine kDa) and type II (seven kDa), which were distinct both in terms of primary sequence identity and the disulfide bond pattern (Douliez et al., 2001b).

Plant nsLTPs are ubiquitous proteins encoded by multigene families, as reported in different phylogenetic studies. However, these studies involve a limited number of sequences and/or species: a total of 15 nsLTPs identified in Arabidopsis (Arondel et al., 2000), restricted to Poaceae (Jang et al., 2007), Solanaceae (Liu et al., 2010; D’Agostino et al., 2019), or Gossypium (Li et al., 2016; Meng et al., 2018). Recently 189 nsLtp genes were identified in three Gossypium species (Li et al., 2016) and 138 nsLtp genes in the single Gossypium hirsutum species (Meng et al., 2018) using traditional sequence homology approaches. As for Solanum lycopersicum, D’Agostino and collaborators identified 64 nsLtp gene sequences using an Hidden Markov Model approach (D’Agostino et al., 2019). Around 200 nsLTPs have been identified in wheat, rice and Arabidopsis genomes and classified into nine different types according to sequence similarity (Boutrot, Chantret & Gautier, 2008). More extensive studies including ancestral plants indicate that nsLTPs are also present in liverworts, mosses and ferns, but not present in algae (Edstam et al., 2011; Wang et al., 2012a). However, the efforts made so far to classify the members of the nsLTP superfamily were including proteins that do not satisfy the strict criteria of 8CM pattern (Edstam et al., 2011; Wang et al., 2012b). In comparison to previous studies, we computed the most extensive phylogenetic analysis, sampling 797 nsLTP sequences from 123 different species. This allows to enrich the phylogenic tree of many evolutionary events that would have been hidden with more restrictive species choices. These events are essentially gene duplications and have a major influence on the family evolution that could be correlated to three-dimensional (3D) structure evolution.

From a structural point of view, the nsLTP family belongs to the all-alpha class in the SCOP database (Murzin et al., 1995), as these small proteins contain four or five helices organized in a right-handed superhelix. To date, only 30 3D redundant structures corresponding to eight different proteins have been experimentally determined. According to SCOP, the protein fold called “Bifunctional inhibitor/lipid-transfer protein/seed storage 2S albumin” is found in at least six distinct plant nsLTPs for which the 3D structure has been solved (from five species Triticum aestivum, Hordeum vulgare, Zea mays, Oryza sativa and Triticum turgidum), and one soybean hydrophobic protein. In the RCSB Protein Database (Berman et al., 2000) we listed four more plant nsLTP 3D structures (from Nicotiana tabacum, Phaseolus aureus, Prunus persica and Arabidopsis thaliana). According to the CATH database (Orengo et al., 1997), nsLTPs belong to the “Mainly alpha” class. They display the “Orthogonal Bundle” architecture and the “Hydrophobic Seed Protein” topology. At this level, only one homologous superfamily called “Plant lipid-transfer and hydrophobic proteins” can be found. The superfamily appears to contain 10 distinct protein sequences, lacking the Arabidopsis thaliana nsLTP, but including the soybean hydrophobic protein found in the SCOP database. Of the known nsLTP 3D structures, only Boutrot’s type I, II and IV are represented. An interesting point to be noted is that two different cysteine pairing patterns have been observed (which correspond to a single cysteine switch between two disulfide bridges): C1–C6 and C5–C8 in type I structures; C1–C5 and C6–C8 in type II and IV structures. However, C2–C7 and C3–C4 bridges are common to all known nsLTP structures and the overall fold is conserved among the whole family.

From a functional point of view, plant nsLTPs are classified into different families depending on the scope of interest and their properties (Liu et al., 2015). Plant nsLTPs belong to the Prolamin superfamily (AF050), which includes the largest number of allergens (Radauer et al., 2008). Indeed, several nsLTPs from fruits of the Rosaceae family, nuts or different vegetables are food allergens, with fruit nsLTPs being mainly localized in the peel (Salcedo et al., 2007; D’Agostino et al., 2019).

Plant nsLTPs are members of the pathogenesis-related proteins and compose the PR14 family (Van Loon, Rep & Pieterse, 2006). Their role in plant defense mechanisms has been shown by the induction of nsLtp gene expression following pathogen infections, overexpression in transgenic plants, or their antimicrobial properties (Molina & Garcia-Olmedo, 1993; Cammue et al., 1995; Li et al., 2003; Girault et al., 2008; Sun et al., 2008). A role in plant defense signaling pathways has also been suggested for an Arabidopsis type IV nsLTP, which needs to form a complex with glycerol-3-phosphate for its translocation and induction of systemic acquired resistance (Maldonado et al., 2002; Chanda et al., 2011). One wheat nsLTP competes with a fungal cryptogein receptor in tobacco plasma membranes and, when the LTP is complexed with lipids, its interaction with the membrane and its defense activity are enhanced (Buhot et al., 2001, 2004). In wheat, nsLtp genes display a complex expression pattern during the development of the seed (Boutrot et al., 2005). NsLTPs may also be involved in plant defense mechanisms through their participation in cuticle synthesis (DeBono et al., 2009). This function is supported by their extracellular localization (Thoma, Kaneko & Somerville, 1993; Pyee, Yu & Kolattukudy, 1994), the expression of different nsLtp genes in leaf epidermis (Sterk et al., 1991; Pyee & Kolattukudy, 1995; Clark & Bohnert, 1999), a positive correlation between nsLtp gene expression and cuticular wax deposition (Cameron, Teece & Smart, 2006), and their ability to bind cutin monomers (i.e., hydroxylated fatty acids) (Douliez et al., 2001a). In addition, nsLtp gene transcripts are abundant or specifically present in trichomes and one tobacco nsLTP seems to be required for lipid secretion from glandular trichomes indicating that nsLTPs may play a role either in the secretion of essential oils or in defense mechanism (Lange et al., 2000; Aziz et al., 2005; Choi et al., 2012). Several nsLtp genes are up or down-regulated by application of different abiotic stresses including low temperature, drought, salinity and wounding (Wang et al., 2012a; Treviño & O’Connell, 1998; Gaudet et al., 2003; Maghuly et al., 2009). A cabbage nsLTP isolated from leaves stabilizes thylakoid membranes during freezing (Sror et al., 2003). Transgenic orchids transformed with a rice nsLTP exhibit an enhanced tolerance to cold stress (Qin et al., 2011).

Function in male reproductive tissues has also been shown for a lily nsLTP involved in pollen tube adhesion (Mollet et al., 2000; Park et al., 2000) and the Arabidopsis LTP5 implicated in pollen tube guidance in the pistil (Chae et al., 2009; Chae & Lord, 2011). A tobacco nsLTP that accumulates in pistils has been shown to be involved in cell wall loosening, and this activity relies on the hydrophobic cavity of the protein (Nieuwland et al., 2005).

Non-specific LTPs are possibly involved in a range of other biological processes, but their physiological functions are not clearly understood. An analysis of gain of function or defective plant mutants can address these issues (Maldonado et al., 2002; Chae et al., 2009). Site directed mutagenesis (SDM) led to the identification of residues involved in their antifungal activity, lipid binding and lipid transfer (Ge et al., 2003; Cheng et al., 2008; Sawano et al., 2008). However, even if they remain extremely valuable, these approaches are time-consuming and have so far been limited to a small number of proteins. They need to be computationally assisted, using information emerging from big datasets.

Current bioinformatics programs such as GeneSilico Metaserver (Kurowski & Bujnicki, 2003) or MESSA (Cong & Grishin, 2012) provide an overview of known information about protein sequences, structures and functions. However, studying inner relationships into such complex superfamilies of proteins as the nsLTP superfamily requires a knowledge visualization and classification tool that still needs to be developed.

This paper describes both the development of new tools together with the use of these tools to improve our comprehension of the nsLTP superfamily.

Materials and Methods

nsLTP sequences

Definition of the protein sequence set

A first pool of plant nsLTPs sequences was retrieved from the UniProtKB (Swiss-Prot + TrEMBL) (http://www.uniprot.org), Phytozome (http://www.phytozome.net) and NCBI databases (http://www.ncbi.nlm.nih.gov), using either Blast or keyword queries (“Plant lipid transfer protein,” “viridiplantae lipid transfer protein,” “plant A9 protein,” “A9 like protein,” “tapetum specific protein,” “tapetum specific,” “anther specific protein,” “A9 Fil1”). Original data obtained on the Theobroma cacao genome were also investigated (Argout et al., 2011). From this large pool of proteins, the plant nsLTP dataset was defined according to a new set of criteria: (i) sequences from 60 to 150 residues long, including signal peptide; (ii) containing strictly eight cysteine residues after removal of the signal peptide; (iii) cysteine residues distributed in the 8CM pattern (C-Xn-C-Xn-CC-Xn-CXC-Xn-C-Xn-C). We excluded multi-domain proteins, that is, the hybrid proline-rich and hybrid glycine-rich proteins in which the signal peptide is followed by a proline-rich or a glycine-rich domain of variable length (José-Estanyol, Gomis-Rüth & Puigdomènech, 2004). For each sequence, the signal peptide was detected and removed using SignalP 3.0 (Bendtsen et al., 2004). In all, including the wheat, rice and Arabidopsis sequences previously identified by Boutrot, Chantret & Gautier (2008), 797 non-redundant mature amino acid sequences belonging to more than 120 plant species were kept for analysis. This dataset is available online (DOI 10.18167/DVN1/UNKLA6).

Sequence alignments and phylogenetic analysis

In order to achieve the best alignment, the pool of 797 sequences was aligned using both the MAFFT (Katoh et al., 2002; Katoh & Toh, 2010) and MUSCLE (Edgar, 2004) programs with respective parameters of 1.53 for gap opening, 0.123 for gap extension and BLOSUM62 matrix; maximum iteration 16.

The two resulting multiple alignments were compared and conflicts between the two were highlighted. To discriminate between the two different cysteine patterns suggested (see Results section), a restricted analysis was carried out using only the 10 nsLTPs for which at least one structure had previously been experimentally determined. Two new 10-sequence alignments were calculated, one by MUSCLE and one by MAFFT. Using the ViTo program (Catherinot & Labesse, 2004), each alignment was projected on type I, II and IV nsLTP 3D structures, and the spatial distance of equivalent cysteine residues was evaluated. The alignment that minimized these distances was selected as the best one.

Based on the best alignment, a phylogenetic tree was calculated using PhyML (Guindon et al., 2010). Lastly, the tree was reconciled with the overall species tree using the Rap-Green program (Dufayard et al., 2005).

nsLTP three-dimensional structures

Three-dimensional structure modeling

For 10 out of the 797 nsLTP dataset, one or more experimentally determined 3D structures were available and downloaded from the Protein Data Bank (http://www.rcsb.org/pdb). Theoretical structures were calculated for the other 787 proteins using the @tome2 suite of programs to perform homology modeling (Pons & Labesse, 2009) (http://atome.cbs.cnrs.fr). The quality of each final structure model was evaluated using Qmean (Benkert, Tosatto & Schomburg, 2008). Structures with low quality (i.e., for which the cysteine scaffold could not be fully modeled) were discarded from further analysis (see Table 1).

Table 1 Qmean scores obtained by the 797 theoretical models of nsLTPs of this study.

Qmean score (Q)	Nb. of models	Dataset proportion	
Q < 0.2	0	0%	
0.2 < Q < 0.3	1	0.1%	
0.3 < Q < 0.4	30	4%	
0.4 < Q < 0.5	173	22%	
0.5 < Q < 0.6	309	39%	
0.6 < Q < 0.7	221	28%	
0.7 < Q < 0.8	43	5%	
0.8 < Q < 0.9	9	1%	
0.9 < Q	1	0.1%	
Note:

Models obtained by @tome2 present an overall good quality as shown in Table 1 that summarizes the Qmean scores. For 95%, 85% of the models, Qmean scores are above 0.4% and 57% of the models obtained scores ranging from 0.5 to 0.9, which correspond to scores for high-resolution proteins. It is known that disordered protein regions are very flexible regions. While submitted to automatic evaluation, these flexible regions will be considered as regions of bad quality modeling, leading to lower Qmean scores (Benkert, Tosatto & Schomburg, 2008; Benkert, Biasini & Schwede, 2011). Small proteins tend to have lower scores than larger proteins, because of the lower proportion of secondary structures compared to random coils. However, the set of theoretical models calculated by @tome2 obtained overall good Qmean scores.NB: for 121 theoretical structures, the polypeptide chain could not be fully built and the resulting models were lacking at least one of the eight cysteine residues. Such models were discarded and a new pool of 677 structures was retained for further analysis. The models are available at: http://atome.cbs.cnrs.fr/AT2B/SERVER/LTP.html.

Structural classification

All the remaining good-quality theoretical structures, together with the 10 experimental structures composed the 3D structure pool of the study. Except for the cysteine pattern analysis by ViTo, this structural pool was used in all further structural analysis.

The structures were compared to each other in a sequence-independent manner, using the similarity matching method of the MAMMOTH program (Ortiz, Strauss & Olmea, 2002). The RMSD was calculated for each pair of structures, using the superposition between matched pairs that resulted in the lowest RMSD value. This superposition was computed using the Kabsch rotation matrix (Kabsch, 1976, 1978) implemented in the MaxCluster program (Herbert, 2019). We used the RMSD score matrix calculated by MaxCluster as input for the FastME program (Desper & Gascuel, 2002) to calculate a structural distance tree.

nsLTP functional annotation

Extensive bibliographic work was carried out to collect and classify functional information available in the literature about the nsLTPs of the dataset. Gene ontology, plant ontology and trait ontology terms were collected from the Gramene Ontologies Database (http://www.gramene.org/plant_ontology) and organized in a dedicated database, together with the bibliographic references when available. The database was also enriched with additional information, such as methods used for gene expression studies (northern, RT-PCR or microarray data, in situ hybridization), protein purification, in vitro or in planta antifungal and antibacterial activity, lipid binding or transport (fluorescence binding assay or in vitro lipid transfer). Information about tissues and organs used in cDNA libraries was collected from the NCBI databases (http://www.ncbi.nlm.nih.gov). A dataset with all the annotations is available online (DOI 10.18167/DVN1/1O5UAK).

Integrative method 1: sequence–structure–function

This method seeks to identify common ligand binding properties in nsLTPs clustered by sequence similarity.

Sequence consensus for each nsLTP type

A total of 797 nsLTP sequences were clustered by type on the basis of regular expressions derived from the consensus motifs described in (Boutrot, Chantret & Gautier, 2008). Each type subfamily was then aligned individually and the resulting sequence profiles were globally aligned using MUSCLE. For each type subfamily, the most frequent amino acids were selected at each alignment position to build the consensus sequence. A consensus amino acid was replaced by a gap if more than half of the sequences were aligned with a deletion at the considered position.

nsLTP sequence–structure analysis using frequently aligned symbol tree

An original tool was designed to highlight conserved amino acid positions specific to each nsLTP phylogenetic type, and which might be decisive for their function. The algorithm relied on a statistical analysis of each alignment row, after the sequences had been clustered according to their phylogenetic distances.

For each type subfamily, the most frequent amino acids were selected at each alignment position to build the consensus sequence. A consensus amino acid was replaced by a gap if more than half of the sequences were aligned with a deletion at the considered position. We then calculated the amino acid conservations and specificities over each column of the multiple sequence alignment to delineate the functionally important residues in each nsLTP subfamily. This statistical analysis is explained in Fig. 3.

In order to visualize the conserved and divergent regions of the sequences, different color ranges were assigned to the nsLTP phylogenetic subfamilies. Conserved amino acid positions along the whole alignment (conserved core positions) are represented in gray/black, while specifically conserved positions among proteins of the same subfamily (specificity determining positions) are represented in saturated colors corresponding to the family ones. The tool enabled scrolling down of the alignment to easily identify both types of conserved positions and two distant parts of the alignment could be displayed together to compare distant phylogenetic subfamilies.

Contacts with ligands, solvent accessibility and other parameters could also be displayed above the alignment. Using the Jmol interface, conserved amino acid residues could be projected on nsLTP representative 3D structures, so that the potential role of each position could be interpreted geometrically.

Integrative method 2: function–structure–sequence

Structural Trace Display is a method, based on evolutionary trace display (ETD, Erdin et al., 2010), that seeks to identify common structural (1D, 3D) properties in nsLTPs sharing similar functions.

Clustering of the structure tree

As in a phylogenetic tree, nsLTPs in the structure tree were clustered according to their similarity. In the case of this particular tree, the similarity between nsLTPs was measured by a spatial distance in angströms (see paragraph 2/nsLTP 3D structures/Structural classification). Decreasing distance cutoffs ranged from 11.5 Å (one cluster containing all nsLTP structures) to 0.5 Å. Each cutoff application caused a division of the tree into one or more sub-trees that contained leaves (i.e., nsLTP structures) whose structural proximity altogether (represented by the pairwise RMSDs) was up to the value of the applied cutoff.

InTreeGreat: an integrative tree visualization tool

We developed an integrative tree visualization tool called InTreeGreat in order to display the whole or some parts of either sequence or structure distance trees.

InTreeGreat was implemented using PHP and Javascript, in order to generate and manipulate an SVG graphical object.

The main objective of this tool is to graphically highlight correlations between 3D structures, evolution, functional annotations or any available heterogeneous data. In the context of this study, the interface was able to retrieve information from the nsLTP database to annotate the tree.

InTreeGreat includes functionalities such as tree coloration, fading and collapsing. Heterogeneous data related to sequences (e.g., annotations, nsLTP classification) can be displayed in colored boxes, aligned to the tree.

Cluster selection

Using InTreeGreat to investigate our annotated structure tree, we looked for clusters of nsLTPs sharing the same kind of functional annotations. We focused our attention on one interesting functional role highlighted in several nsLTPs: the implication in plant defense mechanisms against pathogens (bacteria and/or fungus). In order to highlight structure–function relationships, we studied three groups of nsLTPs (see Results section for details): (i) the “defense cluster” (43 proteins, distance cutoff = 1.5 Å); (ii) the cluster containing all type I fold proteins (402 proteins, distance cutoff = 3 Å); (iii) a group manually composed of all type I fold nsLTPs for which a functional role in defense and/or resistance against pathogens had been reported in the literature (28 proteins).

Within each of these three clusters, the protein structure showing the shortest RMS distance from all the others was selected as the representative structure of the cluster for the structural trace calculation.

Structural trace calculation

A structure-based sequence alignment was carried out on the nsLTP structures by Mustang software (Konagurthu et al., 2006).

For each previously selected structural cluster, the corresponding set of protein sequences was extracted from the multiple structural alignment of the nsLTPs. The evolutionary trace (ET) method (Lichtarge, Bourne & Cohen, 1996) was applied: the partial multiple sequence alignment was submitted as input for the ETC program (locally installed, http://mammoth.bcm.tmc.edu/ETserver.html) together with the representative structure of the cluster (selected as described in the previous paragraph).

The ETs based on the structural alignments corresponding to the three nsLTP clusters were then compared to each other. To that end, the 30% top-ranked residues of the defense cluster trace were considered as constitutive of the reference trace (i.e., 27 most conserved amino acid residues) and their ranking and scores in the two other traces were analyzed. The results were compiled in a table and graphically visualized using PyMOL (http://www.pymol.org/).

Results

nsLTP sequences analysis

nsLTP dataset

Over the last four decades numerous proteins, whose ability to transfer lipids has not always been demonstrated, have been annotated as nsLTPs on the basis of sequence similarity. In order to understand more clearly the functional characteristics and the inner variability of this family, we focused the study on the monodomain proteins, which present the strict and only nsLTP domain, that is, the eight-cysteine residues arranged in four disulfide bridges. In total, including the wheat, rice and Arabidopsis sequences previously identified (Boutrot, Chantret & Gautier, 2008), together with sequences from the UniProt (Swiss-Prot/TrEMBL), NCBI and Phytozome databases, 797 non-redundant mature nsLTP sequences belonging to more than 120 plant species were kept for analysis. This first step allowed the selection of a relevant set of proteins covering variability in the nsLTP family. The number of sequences (797) was also large enough to challenge any analysis method we used during this study.

Sequence alignment and cysteine pattern

The alignment of all non-redundant protein sequences for which the 3D structure was experimentally determined (10 sequences) was carried out twice, using the MUSCLE program on the one hand, and the MAFFT program on the other hand. The resulting alignments obtained with standard settings are shown on Figs. 1A and 1B.

Figure 1 Effect of alternate cysteine residue alignments on the superposition of type I and II nsLTP experimentally determined structures.

(A) Common alignment of Cys5 (type I), Cys5′ (types II and IV) (green) and Cys6 (type I), Cys6′ (types II and IV) (magenta) of nsLTP sequences generated by MUSCLE. Only nsLTPs (PDB IDs) with known experimental structures were considered. (C) 3D projection of this alignment leads to an RMSD of 7.32 Å between type I (blue backbone) Cys6 and type II (pink backbone) Cys6′, colorized as in (A). (B) Type I, II and IV nsLTP alignment generated by the MAFFT program, suggesting that type I Cys5 (dark green) corresponds to type II Cys6′ (light green) rather than type II Cys5′. (D) 3D projection of this alignment leads to an RMSD of 2.15 Å between type I Cys5 and type II Cys6′, colorized as in (B). Note that type IV nsLTPs are structurally close to type II nsLTPs.

In both cases, cysteine residues of the 8CM aligned quite well among the three represented types of nsLTPs (types I, II and IV), except for the Cys5-X-Cys6 (CXC) pattern region (where X stands for any amino acid residue). MUSCLE did align type I Cys5 with types II and IV Cys5′, as well as type I Cys6 with types II and IV Cys6′ (Fig. 1A), just as previous studies typically showed (Liu et al., 2010; Silverstein et al., 2007). However, in the alignment carried out by MAFFT (Fig. 1B), Cys5 of type I nsLTPs was equivalent to Cys6′ of type II and IV nsLTPs, and not to the corresponding Cys5′.

To determine which of these two alignments better aligned nsLTP type I and II CXC patterns, we used ViTO program to compare the effects on the 3D structures of both sequence alignments.

The 3D projection of MUSCLE sequence alignment (Fig. 1C) showed that the attempt to spatially superimpose Cys5 and Cys6 of type I nsLTPs with Cys5′ and Cys6′ of type II nsLTPs, respectively, was very expensive in term of Cys C-alpha RMSD value (7.32 Ǻ).

On the contrary, according to the 3D projection of MAFFT sequence alignment (Fig. 1D), type I Cys5 could be well superimposed with type II Cys6′, with a Cys C-alpha RMSD value dropping to 2.15 Å. Furthermore, type I hydrophylic X residue was exposed to the solvent, whereas type II apolar X residue was orientated toward the core of the protein, increasing the stability of the proteins. For these reasons we think that the sequence alignment calculated by MAFFT is more relevant.

This compound approach allowed us to sort the 797 sequences unambiguously into two main families.

nsLTP sequence classification

Our dataset was mainly composed of nsLTPs from angiosperm species (19 monocotyledonous species and 83 eudicotyledonous species) plus five gymnosperm species (35 sequences), one lycophyte species (34 sequences) and two bryophyte species (17 sequences). The monocot sequences were mainly represented by Poales nsLTPs (256 out of 270 sequences) whereas Rosid nsLTPs were the most abundant (364 out of 436 sequences) within eudicots.

The phylogenetic analysis showed that the pool of proteins clustered into nine different types, all highly supported (branch support >0.84). This result mostly confirmed Boutrot’s classification, defined on Arabidopsis thaliana, Triticum aestivum and O. sativa nsLTP sequences, in nine types (Boutrot, Chantret & Gautier, 2008). The main differences were the identification of a new group (named type XI), including 23 sequences, and that Boutrot’s type VII nsLTPs disappeared from our dataset. Indeed, the latter did not satisfy the 8CM criteria as they have only seven cysteine residues in their sequences. For the same reason, Wang’s A, B, C and D types (Wang et al., 2012b) were not represented in our classification.

Type I nsLTPs formed a well-supported monophyletic group (branch support of 0.879) and predominated over the other types, as they accounted for more than half of our dataset (417 out of 797 sequences). This was also observed by Wang et al. (2012b) with a set of 595 nsLTPs. Conversely, in Solanaceae, the most abundant nsLTPs belong to a type referred to as type X by Wang (70 out of 135 sequences) and which seems specific to that plant family (Liu et al., 2010) but was not present in our dataset. To avoid any confusion, we did not used type X denomination in this work. Type II nsLTPs were the second most abundant type (126 sequences) followed by type V (70 sequences) and type VI (60 sequences). Type IX (12 sequences) was mainly composed of Physcomitrella patens nsLTPs and type XI (23 sequences) was mainly composed of nsLTPs from eudicot species. A total of 12 nsLTPs were not included in any of the identified types: these were mainly Physcomitrella patens (six sequences) and Selaginella moellendorfii (four sequences) proteins (Fig. 2).

Figure 2 nsLTP sequence classification.

Dendrogram built on MAFFT alignment of the 797 nsLTP sequences, using Dendroscope program (Huson & Scornavacca, 2012). The different nsLTP types are displayed using various colors and the number of sequences in each type is specified in parenthesis. Branch support values of each group are indicated on the corresponding nodes.

Type XI were grouped in a cluster of 23 sequences in the phylogenetic tree, fairly well supported by a branch of 0.879 aLRT SH-like score. Type XI appeared between type I and the other types, but even though type XI and I were grouped together in the tree, it remained unclear which of the three groups (type I, type XI and other types) diverged first.

All nsLTP types were represented in eudicots while types IX, X (in Wang’s nomenclature) and XI were not identified in monocot species. Within the lycophyte and bryophyte species, no type II, III, IV nor VIII nsLTPs were identified. In the same way, no type III, VIII, IX or XI were identified within gymnosperm species. A total of 10 out of the 16 moss Physcomitrella patens nsLTPs were type IX, the other six remained un-typed, and the only liverwort Marchantia polymorpha nsLTP was a type VI. The 34 Selaginella moellendorfii sequences were mainly types V and VI (15 and 7, respectively) and seven nsLTPs belonged to the new type XI. The Physcomitrella patens and Selaginella moellendorfii nsLTPs formed independent branches or were located at the same branch as type V in Wang’s phylogenetic tree (Wang et al., 2012a) and were included in type D in Edstam’s classification (Edstam et al., 2011). However, Edstam’s type D included rice and Arabidopsis type IV, V and VI nsLTPs. Edstam’s type G nsLTPs, which corresponded to GPI-anchored LTPs and types J and K, which did not fit our molecular mass criteria or contain more than one 8CM motif were not included in our dataset.

According to Yi et al. (2009), Allium nsLTPs may constitute a novel type of nsLTPs harboring a C-terminal pro-peptide localized in endomembrane compartments. In the prolamin superfamily tree of Radauer & Breiteneder (2007), the Allium cepa nsLTP (192_ALLCE) is closed but not included in the type I nsLTPs. In our phylogenetic tree, the three nsLTPs from Allium species were classified as type I. The 501_MEDTR medicago nsLTP was suggested to belong to a new nsLTP subfamily involved in lipid signaling (Pii, Molesini & Pandolfini, 2013) like Arabidopsis DIR1 (151_typeIV_ARATH). In our phylogenetic tree, both proteins were identified as type IV nsLTPs.

The Theobroma cacao genome contains at least 46 nsLtp genes distributed across the 10 chromosomes. Several Theobroma cacao nsLtp genes are organized in clusters, as observed in the rice, Arabidopsis and sorghum genomes (Boutrot, Chantret & Gautier, 2008; Wang et al., 2012b). Apart from nine sequences that were classified in the new type XI, all other Theobroma cacao nsLTPs were classified within the previously identified types and belonged mainly to type I (14 sequences), type VI (seven sequences) and type V (six sequences).

It is worth noting that all the nsLTPs identified as allergens (IgE binding) were type I, except one type II nsLTP (545_BRACM). The 501_MEDTR nsLTP was also suggested to play a role in the root nodulation process (Pii et al., 2009; Pii, Molesini & Pandolfini, 2013). Lipid signaling (lyso-phosphatidylcholine) has been reported to be involved in symbiosis (Bucher, Wegmüller & Drissner, 2009).

This analysis was the most extensive so far and confirmed most of Boutrot’s classification, but complements it due to a larger dataset and a more detailed phylogeny analysis.

nsLTP structure analysis

nsLTP structure modeling

Given the nsLTP fold conservation and the quality of the available experimental structures, reliable models could be obtained for all nsLTPs using the comparative modeling method, although the sequence identity observed among all nsLTP sequences was only in the range of 25%.

Models deduced by fold-recognition using the @TOME-2 server displayed overall good quality, as shown in Table 1 summarizing the Qmean scores. For 96% of the models, Qmean scores were above 0.4, and 57% of the models obtained scores ranging from 0.5 to 0.9, corresponding to scores for high-resolution proteins.

For 121 theoretical structures, the polypeptide chain could not be fully built and the resulting models were lacking at least one of the eight cysteine residues. Such models were discarded and only the complementary pool of 677 structures was kept for further analysis.

All the structural alignments and 3D models are available at: http://atome.cbs.cnrs.fr/AT2B/SERVER/LTP.html

nsLTP sequence–structure relationships

In order to challenge the structure–function relationship analysis on such a big set, we decided to develop a new tool called FAST, which builds consensus sequences for each family, and highlights the sequence conservation and specificities on the alignment and the associated 3D structures.

Figure 3 shows the consensus sequence alignment for all nsLTP types. The pool of 797 sequences was clustered by type on the basis of regular expressions derived from the consensus motifs described by Boutrot, Chantret & Gautier (2008). Each type subfamily was then aligned individually and the resulting sequence profiles were globally aligned using MUSCLE.

Figure 3 Consensus sequence alignment for all nsLTP types.

The indicated amino acids are the most frequent for each type of nsLTP and vertical arrows indicate residues analyzed in detail in the following text. The residues are colored as follows: (1) The sequences were sorted according to the FastME phylogenetic tree order. (2) A color was assigned to each sorted sequence according to a rainbow color gradient order. (3) Let i be a position of the alignment, a(i) be an amino acid at position i and A(a(i),i) be the amino acid cluster to which a(i) belongs and which has the lowest Fisher’s test probability FP(A(a(i),i)) relatively to any tree cluster. The color of each aligned amino acid a(i) is coded using the hue-saturation-value color space: the color hue of a(i) corresponds to the hue of the median sequence containing an amino acid of the cluster A(a(i),i) at position i. The color saturation of a(i) is proportional to the specificity score -log(FP(A(a(i),i))). The color value or intensity of a(i) is reversely proportional to the conservation score of the column i. The more conserved an amino acid cluster is, the darker its color will be, and the more specific to a phylogenetic group an amino acid cluster is, the more saturated its color will be. Consequently, the globally conserved amino acid clusters are highlighted as dark gray or black columns in the sequence alignment while the amino acids cluster specific to a subgroup of related sequences are highlighted by aligned amino acids with same saturated colors. Furthermore, the amino acid specifically conserved in a given protein can then easily be detected by looking at the residues with colors similar to the sequence name color.

Many residues specifically conserved in type I nsLTP1 corresponded to important folding differences between type I nsLTPs on the one side and all other LTP types on the other side. In the following sections, we list type I nsLTP-specific residues whose differential conservation was supported by structural or experimental data.

First, Gly37, which was specifically conserved in type I nsLTPs, allowed very tight contact of helix 1 and helix 2, which were connected by the disulfide bridge Cys17-Cys34. The closest backbone distance between position 13 of helix 1 and position 37 of helix 2 was 3.34 Å in a type I nsLTP structure (PDB code 1mid) while it was 6.45 Å in a type II nsLTP structure (PDB code 1tuk). These increased helix distances closed the ligand tunnel, which was opened in type I nsLTPs between helix 1 and helix 3, and created two distinct cavities separated by a septum in type II nsLTPs (Hoh et al., 2005). Larger distances between helix 1 and helix 2 were predicted in all nsLTP sequences where Gly37 was mutated into larger residues (i.e., all types but I and XI) and should cause major rearrangement of the ligand cavity entrance on this side of the proteins.

Arginine and lysine residues at position 51 and bulky hydrophobic residues at positions 87 and 89 were two other conserved specificities among type I nsLTPs. The side chains at position 51 had type I-specific polar interactions with the ligand at the cavity entrance near the C-terminal loop, which were not found in other nsLTP types, as detailed later in Fig. 4.

Figure 4 Cartoon representation of the crystallographic structures (A) 1mid (type I), (B) 1tuk (type II) and (C) 2rkn (type IV).

The residues are numbered and colored as in the multiple sequence alignment of Fig. 3. The ligands are represented as ball and sticks (carbon in white, oxygen in red). Some determining amino acid side chains are also displayed.

In addition, in type I nsLTPs, the 5th and 6th cysteine residues belonged to helix 3 and were bridged with the first and 8th cysteines, respectively. These two-disulfide bridges tightened both sequence termini to the protein core. Conversely, in types II and IV nsLTPs, the 5th and 6th cysteines showed permuted bridging partners (to 8th and 1st cysteines, respectively). The intermediate residue connecting the 5th and 6th cysteines was exposed to solvent in type I nsLTPs, while it was replaced by a bulky hydrophobic residue interacting with the ligand in the type II and IV nsLTP core at position 54 of the alignment. It was shown by site-directed mutagenesis that the replacement of this intermediate residue by an alanine residue perturbed folding, ligand binding and lipid transfer activity in type II nsLTPs (Cheng et al., 2008). The position 54 in our alignment corresponds to residue 36 in Cheng’s article. In the light of these experiments, it is therefore interesting to note that alanine residues were frequent at position 54 in type I nsLTPs, while larger hydrophobic residues almost always occupied this buried position in other nsLTP types.

The mutation to alanine of the residue at position 63 (residue 45 in Cheng’s article) was also shown experimentally to be destabilizing in type II nsLTPs (Cheng et al., 2008). This position was occupied by large hydrophobic residues in all nsLTPs but types I and V, where alanine residues were frequent, and type III, where it corresponded to a deletion of 12 consecutive residues.

Other residues specifically conserved in type I nsLTPs were helix N-capping Thr6 and Thr47, whose side chains formed stabilizing hydrogen bonds with the protein backbone, and Tyr20, which was the center of a conserved hydrophobic cluster with Pro30 and Leu/Ile79. The interaction of Tyr20 with Pro30 was experimentally confirmed by the large up field shift of Pro30 (Hα, Hδ) protons (Poznanski et al., 1999). This conserved cluster was stabilizing the interface between helices 1 and 4, but did not participate in the ligand cavity. This particular helix interface was also observed in nsLTP types III, VI, VIII and XI.

We then analyzed the atomic interactions observed between type I nsLTPs and their associated ligands in 19 PDB structures ((1fk0, 1fk1, 1fk2, 1fk3, 1fk4, 1fk5, 1fk6, 1fk7, 1mzl, 1uva, 1uvb, 1uvc, 1bv2, 1rzl, 2b5s, 2alg, 1bwo, 1mid, 1t12). Most contacts involved hydrophobic side chains of the type I nsLTP proteins and carbons of the ligands. Marginally, the most frequent polar contacts involved the side chains of conserved arginines at position 46 of the type I nsLTP alignment, lysines at position 54, aspartic acids at position 90, and various polar atoms of histidines, lysines and asparagines at position 37. It should be stressed that none of these polar interactions were shared by more than 31% of the protein-ligand complexes (fewer than 6/19 PDB structures) although the least similar protein pair from the 19 structure set shared 67% sequence identities. This low level of polar contact conservation in homologous proteins with very similar sequences clearly indicated that no specific polar interactions anchored the protein-ligand complexes in particular conformations. From this statistical analysis of protein-ligand polar contacts that did not exhibit a preferential cavity region for the interaction with the ligand polar heads, it could not be concluded that there was a preferred ligand orientation in the type I nsLTP tunnel. This observation was supported by recent protein-docking simulations and protein binding evaluations, which also concluded on a lack of preferred orientations of the ligand in the cavities of type I nsLTPs, and clear dominance of hydrophobic interactions in the protein-ligand interface (Pacios et al., 2012).

Lastly, positions 82 to 94, which corresponded to the C-terminal loop, included some more residues specifically conserved in nsLTPs. This loop was much longer in type I nsLTPs than in other types, and had a major impact on the orientation of the ligand in the cavity, as shown in Fig. 4.

Conserved and specific residues in the nsLTP family

The potential impact of variability within the nsLTP family on the tree dimensional structure of the proteins was further investigated. As shown in Fig. 4, the ligand cavity opening near the C-Terminal loop was very different when we compared the nsLTP structures of type I vs. those of types II and IV. The C-terminal loops connected the 4 helices to the 3 helices through the disulfide bridge between cysteine residues localized at alignment positions 95 and 55. Both helices 2 and 3 and the C-terminal loop were longer in type I than in types II and IV nsLTPs. In the type I nsLTPs, these elongations created a ligand cavity entrance along an axis perpendicular to the figure plane, while in types II and IV nsLTPs, the entrance was approximately parallel with the figure plane. Consequently, ligands would access the cavities on opposite sides of the C-terminal loop in type I vs. types II and IV nsLTPs. Helix 2 and 3 were extended by an extra turn in type I nsLTPs comparatively to the structures of the other types. Moreover, the small space left in between helices 2 and 3 and the C-term loop was capped in types II and IV by bulky hydrophobic residues (Phe54 in 1tuk and Phe51 in 2rkn), while that position was occupied by a positively charged lysine or arginine in type I nsLTPs (red colored Arg51 in 1mid), whose side chain formed a hydrogen bond with the polar tail of the ligand.

The structural differences observed between type I nsLTPs vs. types II and IV can be generalized to other nsLTP types by looking at the alignment of consensus sequences in Fig. 3. First, the extension of helices 2 and 3 in type I nsLTPs corresponded to a six to eight residue insertion in the consensus sequence alignment, which differentiated type I from every other type of nsLTPs. Secondly, the C-terminal loop connecting the last two cysteine residues was, on average, 13 residues long in type I nsLTPs, while this loop was shortened to 6, 6, 7, 12, 9, 8, 6 and 9 residues long in types II, III, IV, V, VI, VIII, IX and XI, respectively. Lastly, the capping hydrophic residues at positions 54 and 51 of types II and IV nsLTPs were also observed in all the other nsLTP types. These conserved differences between type I and other types of nsLTP sequences indicated with high confidence that the global fold of type I LTP differed from the fold of the other nsLTP types and that the ligand cavity entries in type I nsLTPs were uniquely located.

The fold of type I nsLTPs will be hereafter referred to as “Type-1 fold” and the alternative fold of Types II to XI will be referred to as “Type-2 fold”. (in other words: Roman numerals I to XI correspond to phylogeny analysis while Arabic numerals 1 or 2 refer to structural analysis)

The preceding analysis of the evolutive conservations specific to type I nsLTPs revealed many residues whose role could be explained by local structural differences with the available types II and IV nsLTP structures. This comparative structure analysis confirmed the clear separation between type I and all the other nsLTP types initially observed in the phylogenetic tree inferred from a multiple sequence alignment of the 797 available proteins. The key residues were usually present in type I nsLTPs only and suggested that many structural differences observed when comparing type I vs. types II and IV nsLTPs should also be observed vs. other nsLTP types, particularly regarding ligand orientation and cavity entrances. This observation should guide the choice of templates when nsLTP types with unknown structures are modeled by homology.

Structure classification

In order to correlate the evolution of protein sequences and the impact on the corresponding structures, we produced a circular tree according to structural distances (Fig. 5). Whereas type I remained together in this second classification, other phylogenetic types were relatively scattered in the tree. A majority of type II nsLTPs remained together in this tree, as was also the case for type IV and type III, but no clear and reliable segregation between all non-type I nsLTPs could be made. Looking at the 3D structures allowed us to confirm the hypothesis that only two major structural types could be distinguished. They will be hereafter, referred to as “Type-1 fold” and “Type-2 fold”.

Figure 5 nsLTP structure classification.

Dendrogram built on Mustang structure-based sequence alignment of the 727 nsLTPs for which a reliable 3D model has been calculated. The two main fold types are displayed in red (Type-1 fold) and black (Type-2 fold). In order to study their distribution in term of structural families, nsLTP structures are colored according to the previously determined phylogenetic type they belong to (same colors as used in Fig. 2). Phylogenetic type I nsLTPs display the Type-1 fold and all other nsLTPs follow the Type-2 fold.

Several studies also showed that type I and type II nsLTPs differed through the characteristics of the residue standing between Cys5 and Cys6, being, respectively, hydrophilic in type I and apolar in type II proteins (Douliez et al., 2001b; Marion et al., 2003). Based on the multiple sequence alignment of the 797 nsLTPs and observation of the nature of the central residue in the CXC pattern, together with the observations made in the preceding sequence–structure analysis, we suggest that types III, IV, V, VI, VIII, IX and XI nsLTP C5 and C6 residues will adopt the same spatial conformation as type II proteins, that is, the “Type-2 fold”.

nsLTP structure-function relationship

Dealing with big datasets can be cumbersome and requires a very efficient interface. To address this challenge, we developed InTreeGreat, a Javascript/PHP interface, compatible with every standard web navigator. It is able to display and explore any tree and to deal with branch and leaf coloring, branch lengths, branch support (or any other branch labels), and can aggregate heterogeneous data (annotations, expression profiles, etc.). Figure 6 shows how InTreeGreat can be used to display phylogenic trees together with various types of annotations.

Figure 6 InTreeGreat view of the structure tree.

The left pane shows the phylogenic tree of the nsLTP structures colored according to type and the right pane represents a close-up of the Type I (colored in red) part of the tree. For clarity, some sub tree parts for which no annotation was available have been collapsed. They appear as gray triangle and the number of structures they contain is indicated. NsLTPs for which a functional annotation is available are highlighted with a gray box in the left column. On the right side of the tree, several columns appear that correspond to annotations (PO, GO), number of leaves in a collapsed sub-tree together with colored boxes. The first column of boxes shows alternative colors to enhance the clusters, the other ones correspond to each keyword selected among the annotations of the database (here: “defense” or “resistance”). Keywords “defense” or “resistance” used in functional annotation are highlighted with a colored box (blue and red, respectively). The “defense cluster” (see text) has been enlarged (black border) for a better view.

Among the annotated nsLTPs (433 out of 797), we focused on those that had been reported for their role in plant defense and/or resistance against pathogens (bacteria and/or fungi). To simplify, we shall hereafter refer to them as “defense nsLTPs” in the present discussion. By investigating structural similarities between the 31 identified defense nsLTPs in our annotated dataset, we attempted to identify key amino acid residues that would be good candidates for SDM experiments as they may bestow their functional properties on these proteins.

Looking at the distribution of the defense nsLTPs in our structural classification (Fig. 6) we observed that they were predominantly found in the type I part of the tree (28 proteins), with only three defense nsLTPs with a type II (85, 151, 501—UniProtKB—P82900: Non-specific lipid-transfer protein 2G, Q8W453: Putative lipid-transfer protein DIR1, O24101: Lipid transfer protein). We therefore preferred to focus on the Type-1 fold nsLTPs and study the structural trace(s) inside this important subfamily of nsLTPs.

The cluster containing all Type-1 fold defense nsLTPs corresponded to the whole type I part of the tree (402 members). The corresponding structural trace was calculated, but it could not be linked to the defense function, as the proportion of annotated nsLTPs with a defense function was too low (28 out of 402, i.e., 7%).

In order to obtain a meaningful trace of the potential defense function, we needed to select a cluster with a higher proportion of annotated defense nsLTPs. The best cluster we could find was a relatively small cluster (43 members) of proteins with a structural distance no greater than 1.5 Å (i.e., 1.5 cut off), which contained 33% of the defense nsLTPs (i.e., 10 out of 31 proteins). This cluster will be referred to as “defense cluster” in the further discussion.

The structural trace of the defense cluster showed several differences in comparison with the structural trace of the Type-1 fold cluster (Table 2). Apart from the eight Cys residues that were common to all nsLTPs, the 30% top ranked (i.e., 27 residues) most conserved residues were not the same, or did not come in the same order in both traces. According to the defense cluster trace, residue Asp at position 259 of the alignment (Asp45 in protein 525) was as strongly conserved as the eight Cys residues. Residue Ile at position 402 (Ile80 in protein 525) was among the four best ranked residues after the eight Cys residues and obtained a very low coverage, variability and rvET score. In terms of the ranking of these two (amino acid) residues in the Type-1 fold nsLTP trace, they appeared to occur much later in the ranking (20th and 21st rank, respectively) with much higher rvET scores and large variability in terms of the number and physico-chemical properties of the residues (Table 2). It can be suggested that these two residues were not critical for maintaining structure integrity, but could bestow functional specificity on the proteins classified in the defense cluster. In the trace obtained for the group composed by all the other Type-1 fold defense nsLTPs, both residues Asp and Ile were among the four best ranked residues after the eight Cys residues and also showed good coverage and rvEt scores (Table 2).

Table 2 Compared analysis of evolutionary trace of three groups of nsLTPs.

A	
Defense cluster (ref. prot. = 525)	
Rank	Residue number	Alignment position	Residue	Coverage	Variability	rvET score	
1	4	93	C	0.10000	C	1.00	
1	14	159	C	0.10000	C	1.00	
1	29	228	C	0.10000	C	1.00	
1	30	229	C	0.10000	C	1.00	
1	45	259	D	0.10000	D	1.00	
1	50	275	C	0.10000	C	1.00	
1	52	277	C	0.10000	C	1.00	
1	72	372	C	0.10000	C	1.00	
1	86	432	C	0.10000	C	1.00	
10	7	137	V	0.13333	AV	1.11	
11	32	231	G	0.13333	SG	1.11	
12	80	402	I	0.13333	VI	1.11	
13	69	367	P	0.14444	PA	1.17	
14	36	236	L	0.15556	LV	1.28	
15	17	165	Y	0.16667	FY	1.59	
16	74	374	V	0.17778	LVIA	1.75	
17	11	154	L	0.18889	LV	1.83	
18	54	289	K	0.20000	VKQ	1.93	
19	65	360	A	0.21111	TALV	2.01	
20	40	247	A	0.22222	TAV	2.13	
21	1	63	A	0.23333	.AD	2.15	
22	33	232	V	0.24444	AVI	2.29	
23	68	364	I	0.25556	LI	2.50	
24	43	256	T	0.26667	TPMAS	2.61	
25	61	344	N	0.27778	KNSV	2.65	
26	47	268	Q	0.28889	RQK	2.71	
27	46	266	K	0.30000	RK	2.75	
B	
Fold 1 nsLTPs (ref. prot. = 437)	
Rank	Residue number	Alignment position	Residue	Coverage	Variability	rvET score	
1	14	159	C	0.05376	C	1.00	
1	29	228	C	0.05376	C	1.00	
1	30	229	C	0.05376	C	1.00	
1	50	275	C	0.05376	C	1.00	
1	52	277	C	0.05376	C	1.00	
6	75	372	C	0.06452	CR	1.75	
7	4	93	C	0.07527	CA	3.00	
8	89	432	C	0.08602	CDN	4.36	
9	72	367	P	0.09677	PASLQG	7.27	
10	46	266	R	0.10753	RKTAPIQD	11.55	
11	7	137	V	0.11828	VALISGT	11.81	
12	32	231	G	0.12903	GSAEQVHR	13.26	
13	36	236	L	0.13978	LVIM	13.58	
14	77	374	V	0.15054	VLTAINP	13.66	
15	17	165	Y	0.16129	YFAH	13.82	
16	40	247	A	0.17204	ATSVIRPL	14.49	
17	68	360	A	0.18280	ATVLFIM	14.52	
18	71	364	I	0.19355	LIVTAPFM	14.53	
19	54	289	K	0.20430	KVQIERLMHTS	15.40	
20	45	259	D	0.21505	DAENITLRG.K	15.74	
21	83	402	I	0.22581	IVFPLTAKW	15.92	
29	33	232	V	0.31183	VAILSM	21.38	
32	47	268	R	0.34409	KQRVEMIYSH	24.45	
34	11	154	I	0.36559	VLMIFATP	25.38	
42	64	344	N	0.45161	NGKQDASTLERVFYI	54.16	
56	43	256	T	0.60215	TAPGRSQKDHVMI.LFY	38.13	
61	1	63	A	0.65591	AHETDVPSGFQL	39.96	
C	
Defense nsLTPs outside cluster (ref. prot. = 525)	
Rank	Residue number	Alignment position	Residue	Coverage	Variability	rvET score	
1	4	93	C	0.11111	C	1.00	
1	14	159	C	0.11111	C	1.00	
1	29	228	C	0.11111	C	1.00	
1	30	229	C	0.11111	C	1.00	
1	50	275	C	0.11111	C	1.00	
1	52	277	C	0.11111	C	1.00	
1	72	372	C	0.11111	C	1.00	
1	86	432	C	0.11111	C	1.00	
1	7	137	V	0.11111	V	1.00	
1	69	367	P	0.11111	P	1.00	
11	45	259	D	0.13333	DL	1.15	
12	80	402	I	0.13333	IW	1.15	
13	74	374	V	0.15556	VIN	1.39	
16	17	165	Y	0.18889	YF	1.67	
17	36	236	L	0.18889	LI	1.67	
18	32	231	G	0.20000	GAV	1.76	
20	54	289	K	0.22222	KVQ	1.93	
22	65	360	A	0.24444	AVF	2.04	
23	40	247	A	0.25556	ATVS	2.05	
25	33	232	V	0.27778	VALI	2.59	
27	61	344	N	0.30000	NVDR	2.88	
30	46	266	K	0.33333	KR	3.25	
31	11	154	L	0.34444	LIVM	3.26	
36	43	256	T	0.40000	TPQRS	3.72	
38	68	364	I	0.42222	ILV	3.95	
44	47	268	Q	0.48889	QRK	4.61	
45	1	63	A	0.50000	A.QV	4.63	
Note:

Compared analysis of evolutionary trace of three groups of nsLTPs: (A) the defense cluster (43 proteins), (B) the cluster containing all Type-1 fold nsLTPs (402 proteins) and (C) a group composed by all Type-1 fold defense/resistance nsLTPs, including those which do not belong to the defense cluster (28 proteins). This table lists the 30% top-ranked (=most conserved) residues identified in the defense cluster trace and shows, by comparison, the ranking of these same residues in the other two traces, together with their coverage, variability and rvET score. Residue positions in the reference proteins and in the structure-based sequence alignment are also indicated. Alignment position is the same in all three groups because all three alignments used to perform the traces are extracted from the general multiple alignments of all 797 nsLTPs of the study. Five residues are highlighted for they are differently conserved in the three clusters of proteins (see text).

Three other residues located at positions 137, 154 and 266 of the structural alignment were differently conserved in the three clusters. Interestingly, these three positions showed good conservation ranking, but the variability of the three corresponding residues was notably higher in the Type-1 fold cluster. Indeed, in the defense cluster trace, position 137 was occupied either by a valine or by an alanine residue (Val7 in protein 525) and position 154 was occupied either by a leucine or by a valine residue (Leu11 in protein 525). Thus, both positions were occupied by hydrophobic residues in defense proteins, which was not always the case in Type-1 fold proteins (Table 2). In the same way, position 266 was occupied either by an arginine or a lysine residue (both positively charged residues) (Lys46 in protein 525) in defense proteins, but allowed greater variability in terms of physico-chemical properties in the other proteins harboring a Type-1 fold.

The fact that these three positions of the structural alignment belonged to the top 30% most conserved among all Type-1 fold nsLTPs suggested their importance in these proteins. However, because the variability at these three positions was very small among defense nsLTPs and because the physico-chemical property was strongly conserved, we suspected that residues located at positions 137, 154 and 266 of the structural alignment might bestow functional specificity, at least in the case of defense/resistance proteins.

Figure 7 shows the five residues highlighted in Table 2 in the 3D structural context of the representative protein of the defense cluster (protein 525). In this protein, conserved residues Asp and Ile were located at positions 45 and 80, respectively. The two small hydrophobic residues were Val7 and Leu11 and the positively charged residue was Lys46. All five key residues were located around the ligand cavity (Fig. 7), which allowed either guidance or direct contact with the lipid. This observation was consistent with the suggested hypothesis.

Figure 7 Conserved amino acid residues among the so-called defense cluster, displayed on the 3D structure of nsLTP 525, (“LTP,” UniProtKB—Q1KMV1).

The more the residue is conserved in the 3D alignment, the redder its color appears, then orange, yellow and green. Residues with no significative conservation appears in white on the figure. Residues highlighted in Table 2 and which potential functional implication is discussed (see text) are labeled on the figure.

Non-specific lipid transfer proteins sequence–structure analyses using either FAST or STD revealed some key residues or key positions (in type I: Gly37, Arg/Lys51, bulky hydrophobic residues 87 and 89, Ala54, Thr6, Thr47, Tyr20, Pro30, Leu/Ile79, longer C-terminal loop; large hydrophobic residue 63 in types II, III, IV, VI, VIII, IX nsLTPs). The structural trace analysis highlighted other amino acid residues (in type I defense/resistance nsLTPs: Asp45, Ile80, Val/Ala7, Leu/Val11, Arg/Lys46). It is important to note that these two complementary analyses by FAST and STD were not meant to lead to the same kind of conclusions. Indeed, using sequence information projected on the 3D structure, the first method revealed nsLTP-type-specific amino acid residues that could be involved in structure stabilization and/or ligand interaction, given their structural context. The second method however, considered a set of functionally close nsLTPs sharing a very similar structure and highlighted over-representatively conserved amino acid residues that might thus bestow functional specificity on these proteins. These two approaches took inverse directions in the path sequence–structure–function. The “sequence-to-function” method would lead to more precise conclusions if more data about the inner structural mechanisms of lipid binding were available (only a few structures of nsLTP-lipid complexes have been experimentally determined so far). The “function-to-sequence” method would give us a better overview of the range of nsLTP activities if the functional data were not so rare and heterogeneous.

However, we assumed that this combination of approaches (i) allowed structure–sequence analysis for large multigene families, (ii) could reveal structural patterns related to functions that were not revealed so far, as alignments would have been limited to primary sequences only and (iii) allowed a comparison of groups composed of proteins with an evolutionary connection with groups displaying structural similarity.

Discussion

We combined two powerful alignment algorithms (MAFFT and MUSCLE) together with a 3D projection of the impact of alignment on the structure of proteins (VITO). Real-time monitoring of the impact of gap positions and lengths on the resulting 3D model offered the possibility of discriminating between various alignment possibilities. This allowed us to provide definitive insight into the old debate about the CXC pattern and its implication for the structure of LTPs (Douliez et al., 2000). The resulting alignment allowed us to classify unambiguously all 797 sequences in main two nsLTP families.

The phylogenetic analysis was the most extensive to date, including 797 nsLTP sequences. This was a much more complete description than the previous one (195 sequences, Wang et al., 2012a).

This phylogenetic analysis was conducted from a clearly defined dataset: sequences were selected using unambiguous parameters optimizing the quality of the output tree, also considering our 3D structural integration objective. Although GPI-Anchored LTP could have been included in this study, their incomplete homology with other LTPs and the lack of any experimental 3D structure, convinced us not to include them. Thanks to this choice, alignment quality was preserved, and a better-quality 3D structural model are used. This analysis allowed us to classify unambiguously all 797 sequences in the main two nsLTP families, complementing and reinforcing the former classification by Boutrot (Boutrot, Chantret & Gautier, 2008).

The production of more than 600 3D structural models and the collection of numerous functional annotations enabled progress to be made in the study of structure–function relationships of nsLTPs. The re-use of the ETD method in a close and adapted form (STD) led to the identification of amino acids involved in the functional specialization of some nsLTPs.

STD allowed us to highlight amino acids specific to certain functions. One of the limiting points of this analysis remained the publication bias. Indeed, the annotations were not evenly distributed among available sequences, nor was it possible to distinguish between an unsearched function and a function not found. It seemed difficult to propose a solution to circumvent this bias (Douliez et al., 2000).

The structure tree clearly showed that all Type I ns-LTPs adopted the same folding (Type-1 fold), while all the other proteins adopted the second fold (Type-2 fold). This approach seemed very interesting but did not offer the same level of detail and the same analytical power as the phylogenetic approach. This was understandable, because phylogeny compares the different proteins with a much larger number of parameters (site-to-site mutation, classification of sites by mutation rate, use of refined distance matrix, etc.) while the structure tree only uses the RMSD of the structures taken 2 by 2. While this innovative information was very interesting, it could potentially be improved if we had templates from each sub-family for the generation of molecular models (experimental structures are available for Type I, II and IV). Indeed, at this level of analysis, it is conceivable that models obtained from experimental structures for the other types (III, V, VI, VII, VIII, IX and XI) would provide improved models allowing the detection of other key residues.

Conclusions

Plant non-specific lipid transfer proteins constitute a complex family of proteins whose biological functions are far from well understood. However, it has become clear for years that they are of increasing interest for agronomical and nutritional issues.

Experimental approaches are irreplaceable for accessing their inner functional mechanisms. However, such methods are expensive and time-consuming. Furthermore, they produce a large amount of heterogeneous data. For all these reasons, resorting to bioinformatics methods has long become necessary to organize and analyze existing data, and/or model and hypothesize new data.

This paper presented a new methodology based on the combination of either classical or original bioinformatics approaches, using various computational tools to extract information and suggest new hypotheses from a large pool of experimental data about the plant nsLTP superfamily of proteins.

In this paper, we:

(1) suggested a new definition of the nsLTP superfamily, with a set of criteria based on sequence length, sequence composition (e.g., Cys involved in SS bonds) and structure (monodomain);

(2) confirmed and enriched Boutrot’s phylogenetic classification of plant nsLTP sequences;

(3) demonstrated the need for a small shift in the CXC alignment that reflected the existence of two main distinct nsLTP folds;

(4) calculated 666 good quality theoretical 3D structures of nsLTPs;

(5) developed an original alignment tool to detect conserved and specific positions among the different phylogenetic types of nsLTPs;

(6) used the latter tool to reveal some key residues;

(7) suggested a new structure-based classification of the 676 nsLTP structures now available (10 experimental + 666 theoretical), which that allow clustering by structural similarity;

(8) annotated all available information about the function;

(9) developed an original interface allowing quick visualization of several types of annotations on any phylogenetic tree;

(10) revealed, using structural trace analysis, potential specific amino acid residues involved in plant defense and/or resistance against pathogens.

Our work was made more difficult by the problems of annotation bias for which we did not expect a practical solution. However, it seemed that some of our results could be improved if we had additional experimental structures for all types of nsLTPs.

To researchers who may not grasp the importance of the protein structure–function relationships we would like to insist on three main contributions of the methods presented in this work:

- The structural classification agrees with the sequence classification by phylogenetic types.

- The sequence–structure analysis highlights key-residues explaining the specificities of the different folds.

- The structure–function analysis based on the ET of the aligned sequences can show the functional signature of a subfamily of proteins.

Furthermore, the structural dichotomy between type I nsLTPs and all the others may go unnoticed by anyone who would focus on one sequence and who would not conduct a combined analysis on a large set of sequences. We encourage researchers studying nsLTPs to use this approach combining sequence alignment, phylogeny and structural biochemistry. This would enhance the power of the analysis by drawing a connection between primary sequence, 3D structure and function.

More broadly, we consider that this type of combined approach should be favored for any study involving a multigenic family.

This work was supported by the CIRAD—UMR AGAP HPC Data Centre of the South Green Bioinformatics platform.

The authors are thankful to Dr. Franck Molina for his key role at the beginning of this project and all the fruitful and friendly discussions.

We are thankful to Peter Biggins for the careful and critical review of this manuscript.

Additional Information and Declarations

Competing Interests

Author Contributions

Data Availability

The authors declare that they have no competing interests.

Cécile Fleury conceived and designed the experiments, performed the experiments, analyzed the data, prepared figures and/or tables, authored or reviewed drafts of the paper, approved the final draft.

Jérôme Gracy conceived and designed the experiments, analyzed the data, contributed reagents/materials/analysis tools, prepared figures and/or tables, authored or reviewed drafts of the paper, approved the final draft.

Marie-Françoise Gautier conceived and designed the experiments, performed the experiments, analyzed the data, authored or reviewed drafts of the paper, approved the final draft.

Jean-Luc Pons performed the experiments, contributed reagents/materials/analysis tools, approved the final draft, batch molecular modeling.

Jean-François Dufayard conceived and designed the experiments, analyzed the data, contributed reagents/materials/analysis tools, prepared figures and/or tables, authored or reviewed drafts of the paper, approved the final draft.

Gilles Labesse contributed reagents/materials/analysis tools, approved the final draft, model assessment.

Manuel Ruiz conceived and designed the experiments, approved the final draft, bibliographe updates.

Frédéric de Lamotte conceived and designed the experiments, performed the experiments, analyzed the data, contributed reagents/materials/analysis tools, prepared figures and/or tables, authored or reviewed drafts of the paper, approved the final draft.

The following information was supplied regarding data availability:

The code for the tool “InTreeGrate” is available on GitHub: https://github.com/SouthGreenPlatform/rap-green under license (GNU General Public License v3.0).

All the proteins models are available online: http://atome.cbs.cnrs.fr/AT2B/SERVER/LTP.html. A graphic interface allows exploring, selecting and downloading the models. Models are available under license Licence CC BY-NC 4.0.

The fasta file with the 797 protein sequences with all relevant metadata is available on the CIRAD DataVerse:

de Lamotte, Frédéric, 2019, “Functional Annotation for 797 plant ns Lipid Transfer Protein”, DOI 10.18167/DVN1/1O5UAK, CIRAD Dataverse, V1.

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
