# Peer review of "Comprehensive classification of the plant non-specific lipid transfer protein superfamily towards its sequence–structure–function analysis"

_PeerJ, doi:10.7717/peerj.7504_

## Round 0.1 · original submission · Minor Revisions

The reviewers agreed that your contribution is valuable and request minor revisions in a revised version. Besides the reviewer comments, I would appreciate some comments on how scientists outside of the protein science area could benefit from this important work. For example, people obtain and annotate amino acid sequences from plant genomes or transcriptomes, and they findl LTPs in their annotation process. They even model their sequences with Phyre2 or SwissModel, but do not really grasp the importante of protein folds or the concept of structure and function. A perspective for the structurally-inclined bioinformatician would be nice to have for a broader audience.

Reviewer 1 ·

Basic reporting

no comment

Experimental design

The question posed by the authors is clear and well defined and the methods that have been used are adequate.

Validity of the findings

The findings are sound and good enough.
Discussion and Conclusions are well supported and stated.

Additional comments

The manuscript by Fleury et al. is on the analysis and classification of the non-specific
lipid transfer protein superfamily in plants. It is the most comprehensive analysis performed so far with regard to the number of sequences taken into consideration.
The question posed by the authors is clear and well defined and the methods that have been used are adequate. The findings are sound and good enough.
I found the web resource developed by the authors an added value to the paper and I wonder and ask them if they intend to update it in the future with additional sequences form other species.
I recommend publication, but also suggest some minor compulsory revisions to improve the overall quality of the manuscript.

Abstract:

The authors stated that “More than 800 different nsLTP sequences have been
characterized so far” but immediately after they claim “we comprehensively investigated 797 nsLTP protein sequences”. I suggest avoiding this serious inconsistency or at least to explain it.

Introduction:

I would suggest also to briefly and better describe at the beginning of the introduction the efforts made so far for the classification of the members of the superfamily. This is preparatory to a better and understandable reading of the following.

Lines 50-57: Here I suggest that you improve/expand with more details the review of recent literature. For example, the recent manuscript by D’Agostino et al, https://doi.org/10.1038/s41598-018-38301-z (tomato, Solanaceae) should be cited here as well as https://www.frontiersin.org/articles/10.3389/fpls.2018.01285/full and https://www.nature.com/articles/srep38948 (cotton).

Line 58 Please, remove “-“ at the beginning of the paragraph.

Line 78 Please, remove “-“ at the beginning of the paragraph.

Line 82 Please include the reference https://doi.org/10.1038/s41598-018-38301-z

Line 114 Please replace “NsLTPs” with “Non-specific LTPs”

Lines 120-121 this statement is rather risky and does not correspond to reality. There are hundreds of tools out there to analyze protein gene families. Several of them have been also successfully used by the authors to analyse data for this paper.


I suggest that the authors rephrase the last paragraph of the introduction. As it is , it is not clear enough whether the authors intend to describe the methods they developed for data analysis or whether on the basis of those methods they want to carry out the analysis of the nsLTP superfamily in plant.

Results:
Line 303 Please replace ”of sequence homology” with “of sequence similarity”.

Conclusions:
Line 717 Please replace “to bioinformatics methods” with “to bioinformatic methods”

·

Basic reporting

This is a very interesting and timely work. The paper is well written, references are appropriate and figures are well prepared.

Experimental design

The paper is clearly within the Aims and Scope of PeerJ. The research question is well defined and the work is performed to high standards. All methods are well described.

Validity of the findings

no comment

Additional comments

There are only two very minor points I would like to raise.
1. The authors very convincingly analyse and define the Type 1 and Type 2 folds of nsLTP. In the text, they frequently refer to the Type 2 fold as a "so called Type 2". This always sounds a bit like the authors do not fully trust their own analyses. I would strongly encourage the authors to remove this "so called" from their paper.
2. The authors should check the use of Roman and Arabien numerals throughout the manuscript. There were some instances where I had the feeling that this was mixed up.

Reviewer 3 ·

Basic reporting

Authors have tried to decipher the functions of nsLTP(s) by collating and analyzing protein sequences, structures and some reported physiological functions. On the basis of homology modelling of different types, they have classified nsLPTs into two main sub-types and developed two bioinformatics tools for analysis of the dataset. It is a good bioinformatics study and I would recommend for publication if the issues raised in general comments are addressed.

Experimental design

The work is carried out using established computational methods.

Validity of the findings

Findings are validated within the limitations of bioinformatics approaches. It would of course be best if they are experimentally validated. However that would not be in the scope of this article.

Additional comments

1. Line 116-118: The cited reference for the identification of residues involved in their antifungal activity through SDM has not been carried out in original article Ge et al., 2003.
2. Line 308 and 311: In many places the authors have written 797 number of sequences was analyzed and at some place 798.
3. Line 326-335: Explanation for Figure1-A2 and B2, very confusing and not clear.
4. Line 458-471: The cited reference article Cheng et al., 2008, did not carried out SDM of the residues 54 and 63 in their original article, as claimed by the authors.
5. Line: 573-578: Without any SDM data available on antifungal activity, it is very unlikely to suggest any potential residues involved in antifungal activity by doing sequence analysis only.
6. Data for 797/798 sequences extracted, analyzed from different plant sources should also be provided in supplementary file, as mentioned in Boutrot et al., 2008.

---

## Round 0.2 · Minor Revisions

It is a great pleasure to communicate that your revised manuscript is now scientifically accepted in PeerJ. Thanks for attending the comments from the reviewers, and I am sure it will be a reference document in the area of plant LTPs.

Before final acceptance, however, the Section Editor Gerard Lazo has given the following request:

"Though this appears to be a very in-depth study of lipid transfer proteins, there is missing data which is mentioned but not expanded on. Within the methods used there was mention that gene ontology annotations were culled to build meaning to the classification of the sequences and placed in a dedicated database. There is no mention of where this database is and what kind of information can be gained from it. There may be images derived from the database, but in the form that it provided it is of no use to the reader. There is a need for either a pointer to a database where readers can validate the sequences, data, and annotations assigned; or to provide a supplemental file which lists the 797 sequences used in this study. The readers need to have some sort of launching point to be able to validate the data for their own purposes. The manuscript appears very well rounded, but avoids almost every avenue for a reader to become familiar with the information that was studied. Journal manuscripts are often scanned by text-mining software that locates and extracts core data elements, like gene function. Adding standard ontology terms, such as the Gene Ontology (GO, geneontology.org) or others from the OBO foundry (obofoundry.org) can enhance the recognition of your contribution and description. This will also make human curation of literature easier and more accurate. None of this was visible. "

Please address this request, and then the article can be Accepted

---

## Round 0.3 · accepted · Accept

All the comments and suggestions were attended in this revised version. The requested sequences are available in a CSV and XLXS format at
http://dx.doi.org/10.18167/DVN1/1O5UAK

Therefore, the manuscript is accepted in its current form.